# The Prebiotic Activity of a Novel Polysaccharide Extracted from *Huangshui* by Fecal Fermentation *In Vitro*

**DOI:** 10.3390/foods12244406

**Published:** 2023-12-07

**Authors:** Mei Li, Jian Su, Jihong Wu, Dong Zhao, Mingquan Huang, Yanping Lu, Jia Zheng, Hehe Li

**Affiliations:** 1Key Laboratory of Brewing Molecular Engineering of China Light Industry, Beijing Technology and Business University, Beijing 100048, China; lm9810172023@163.com (M.L.); huangmq@th.btbu.edu.cn (M.H.);; 2Key Laboratory of Soild-State Fermentation and Resource Utilization of Sichuan Province/Key Laboratory of Strong Flavor Baijiu Soild-State Fermentation of China Light Industry/Engineering Technology Research Center of Baijiu Brewing Special Grain of China, Wuliangye Yibin Co. Ltd., Yibin 644007, Chinazhengwanqi86@163.com (J.Z.)

**Keywords:** polysaccharide, *Huangshui*, gut microbiota, prebiotic activity, short-chain fatty acids

## Abstract

A novel polysaccharide, HSP80-2, with an average molecular weight of 13.8 kDa, was successfully isolated by the gradient ethanol precipitation (GEP) method from *Huangshui* (HS), the by-product of Chinese Baijiu. It was mainly composed of arabinose, xylose, and glucose with a molar ratio of 4.0:3.1:2.4, which was completely different from the previous reported HS polysaccharides (HSPs). Morphological observations indicated that HSP80-2 exhibited a smooth but uneven fragmented structure. Moreover, HSP80-2 exerted prebiotic activity evaluated by *in vitro* fermentation. Specifically, HSP80-2 was utilized by gut microbiota, and significantly regulated the composition and abundance of beneficial microbiota such as *Phascolarctobacterium*, *Parabacteroides*, and *Bacteroides*. Notably, KEGG pathway enrichment analysis illustrated that HSP80-2 enriched the pathways of amino sugar and nucleotide sugar metabolism (Ko00520), galactose metabolism (ko00052), and the citrate cycle (TCA cycle) (ko00020). Meanwhile, the contents of short-chain fatty acids (SCFAs) mainly including acetic acid, propionic acid, and butyric acid in the HSP80-2 group were remarkably increased, which was closely associated with the growth of *Lachnoclostridium* and *Parabacteroides*. These results showed that HSP80-2 might be used as a potential functional factor to promote human gut health, which further extended the high value utilization of HS.

## 1. Introduction

It is widely acknowledged that various chronic diseases, such as diabetes, obesity and colon cancer, are closely related to human gut microbiota [1]. Individual health is heavily dependent on the gut bacteria, since these symbiotic bacteria provide humans with certain useful metabolites. Therefore, improving gut microbial diversity and richness is a significant focus in the development and implementation of specialized medicinal and functional diets [2]. Diets exert a significant impact on the composition of gut microbiota. Polysaccharides are an integral component of a healthy diet which can regulate the intestinal microflora [3]. In order to study the probiotic properties of polysaccharides, many researchers reproduced the probiotic properties of polysaccharides under anaerobic conditions through *in vitro* fermentation. Numerous studies have demonstrated that the process of polysaccharide fermentation could produce advantageous metabolites, especially short-chain fatty acids (SCFAs) [3,4]. SCFAs are chemical compounds consisting primarily of acetate, propionate, and butyrate, which possess the ability to not only modulate the makeup of the gut microbiota, but also regulate the host’s metabolism [5]. In addition, research has shown that polysaccharides could adjust the structure of the gut microbiota by promoting the proliferation of beneficial bacteria, such as *Lactobacillus* [6] and *Bacteroidetes* [7].

*Huangshui* (HS), as one of the by-products of traditional Chinese Baijiu production, has garnered significant interest in recent years due to its abundance of organic substances and microorganisms [8]. HS polysaccharides (HSPs), which were one of the most important ingredients of HS, were usually obtained through water extraction and the ethanol precipitation method. The ethanol was added to an aqueous solution containing polysaccharides, causing the dehydration of the polysaccharide molecules, and the dehydration process was followed by conformational changes and aggregation, which were induced by the increased strength of intra-molecular hydrogen bonding. It was observed that different concentrations of ethanol could cause the precipitation of polysaccharide fractions with varying average molecular weights. Therefore, the gradient ethanol precipitation (GEP) method was identified as an efficient and expeditious technique to prepare polysaccharide fractions with a high homogeneity [9]. Yet, up to now, there were no reports on using the GEP method to deal with HS and obtained polysaccharides. In terms of biological activity research, previous studies showed that HSPs displayed several significant biological activities, such as antiinflammation, antioxidation immunoregulatory, and protective effect on intestinal barrier damage [9,10,11,12]. However, the impact of HSPs on microbiota and microbial metabolite patterns in the gut, or in other words, the effect on the composition of the intestinal microbiota composition and the ability to metabolize SCFAs, was still unknown.

Consequently, the aim of the present study was to characterize a new polysaccharide and evaluate its regulatory effects on the makeup of the gut microflora and the contents of SCFAs *in vitro*. It would provide a foundation to explore HSPs as potential prebiotics. Main contents were as follows. Firstly, the new polysaccharide, named HSP80-2, was obtained from HS by the GEP method and its structure and morphology were further analyzed. Then, an *in vitro* fecal fermentation model from four healthy individuals was designed to evaluate the role of HSP80-2 on the diversity and richness of intestinal microflora. Finally, changes in the bacterial community structure and SCFAs were estimated using 16S rRNA and gas chromatography, respectively, to evaluate the prebiotic activity of HSP80-2.

## 2. Materials and Methods

### 2.1. Materials and Reagents

The sample of HS was collected from the Sichuan Yibin Wuliangye Group Limited (Yibin, China). T-series dextran standards were purchased from shodex (Tokyo, Japan), and the monosaccharide standards were purchased from Aladdin Holdings Group Co, Ltd. (Beijing, China). Acetic acid, propionic acid, butyric acid, and valeric acid were purchased from Sigma-Aldrich Chemical Co. (St. Louis, MO, USA). All the other chemicals and reagents used were of analytical grade, and were purchased from Sinopharm Chemical Reagent Co., Ltd. (Shanghai, China).

### 2.2. Extraction and Purification of Crude HSP80 (cHSP80)

After 15 min of centrifugation at 8000 rpm, the solid residue in HS was removed. Subsequently, it was deproteinized by treating with the Sevage reagent (CHCl_3_/BuOH = 4:1, *v*/*v*), and the supernatant was obtained by centrifugation (6000 rpm for 20 min). This process was repeated until all free protein was thoroughly removed. As shown in the schematic diagram of the processing process in Figure 1a, a given volume of ethanol with a purity of 100% (*v*/*v*) was added to the supernatant, with the final content of 40% to obtain the precipitate and crude HSP40 (cHSP40), and the supernatant underwent concentration using a vacuum rotary evaporator to remove the ethanol. Similarly, 100% (*v*/*v*) ethanol was added into the resultant supernatant, until the final concentration was 60%, to obtain another precipitate named crude HSP60 (cHSP60), and the supernatant was evaporated to remove the ethanol. Finally, the remaining supernatant was added with 100% (*v*/*v*) ethanol, until the final content was 80%, to obtain the precipitate which passed through dissolution, dialysis, and lyophilization to obtain the crude HSP80 (cHSP80).

In order to obtain polysaccharides with a higher purity and lower molecular weight, which generally had biological activity, cHSP80 was chosen to be further purified in the present study. The specific operations were as follows. In 10 mL of deionized water, 150 mg of cHSP80 was dissolved. The solute was eluted by deionized water and NaCl (0.20 mol/L), respectively, at a rate of 0.2 mL/min using a DEAE-Sepharose FF (2.6 × 20 cm) ion-exchange column chromatography that had reached a state of equilibrium. The two fractions as shown Figure 1b, including HSP80-1 eluted by water and HSP80-2 eluted by 0.2 mol/L NaCl, were collected and subjected to dialysis, respectively, followed by freeze-drying to obtain pure polysaccharides. In the present study, HSP80-2 was selected for subsequent structural analysis and activity research.

### 2.3. Determination of Molecular Weight of HSP80-2

The average molecular weight of HSP80-2 was mensurated by high performance gel permeation chromatography (HPGPC, Shimadzu Corporation, Kyoto, Japan), combined with the TSK gel GMPWXL column (Tosoh Bioscience, Kyoto, Japan) [13]. An LC 20 high performance liquid chromatography pump (Shimadzu Corporation, Kyoto, Japan) equipped with a 7725i manual injector (Rheodyne L.P, Atlanta, GA, USA) was used. The elution solvent used to extract the samples consisted of 0.1% NaNO_3_ and 0.06% NaN_3_ at a rate of 0.6 mL/min. T-series dextrans standards with varying molecular weights of (6.30, 22.0, 49.4, 334.0, and 642.0 kDa) were employed for constructing the standard calibration curve to calculate the average molecular weight of HSP80-2.

### 2.4. Monosaccharide Composition Analysis of HSP80-2

The analysis of HSP80-2 monosaccharide composition had been modified based on previous methods [14]. 1.0 mg of HSP80-2 was dissolved in 3.0 mL of 2.0 mol/L trifluoroacetic acid (TFA) and hydrolyzed for 4 h at 120 °C. After hydrolysis was completed, 200 µL of polysaccharide hydrolysate was blown dry with N_2_ to remove the excess TFA. Then, the desiccated hydrolysate of HSP80-2 and 100 µL of standard monosaccharides (2.0 mmol/L) were mixed with 250 µL of 0.6 mol/L aqueous NaOH and 500 µL of 0.4 mol/L PM, respectively, in methanol solution, which were heated in a water bath at 70 °C for 2 h. After cooling to an ambient temperature, the mixture was neutralized with 0.5 mol/L HCl. The obtained solution was extracted with the same volume of chloroform and water and the sample was subjected to filtration using a membrane with a pore size of 0.45 μm to analysis. The analytical column used was an Agilent Xtimate C_18_ (4.6 × 200 mm × 5 μm). The UV detection wavelength utilized was 250 nm. The composition of elution was acetonitrile and 0.05 mol/L sodium phosphate at a flow rate of 1.0 mL/min at 30 °C and 20 μL was injected.

### 2.5. Atomic Force Microscopy (AFM) Observation of HSP80-2

The 0.1 mg of the polysaccharide sample was dissolved into 10 mg of distilled water and stirred overnight to obtain polysaccharide solution. 5.0 µL of the HSP80-2 solution was dropped on mica surfaces and dried in the air. The cantilever driving frequency was specified as 70 kHz, while the scanning frequency was specified as 1 Hz. The spring constant was stated as 0.4 N/m with AFM (Bruker Dension lcon, Bruker AXS, Billerica, MA, USA).

### 2.6. Scanning Electron Microscope (SEM) Analysis of HSP80-2

The field-emission scanning electron microscopy (SU8020, Hitachi Int. Tokyo, Japan) was utilized to observe the scanning electron micrograph of HSP80-2. The dried HSP80-2 was stuck onto the sample table with conductive tape and sputtered gold film under the vacuum condition, observed at 500 × and 1000× magnifications.

### 2.7. FT-IR Spectroscopy Analyses of HSP80-2

The sample was combined with potassium bromide (KBr) and subsequently compressed into KBr sample discs. The FT-IR spectra of HSP80-2 was obtained by employing a Nicolet iS10 FT-IR spectrometer (Thermo Nicolet Corporation, Waltham, MA, USA). The infrared scanning, in range, spanning from 4000 to 500 cm^−1^ and the resolution was 4 cm^−1^.

### 2.8. In Vitro Fermentation Evaluation of HSP80-2

The *in vitro* fermentation was performed according to the previously described methods with minor modifications [15]. The basal nutrient was prepared as described in the literature, and autoclaved at 121 °C for 20 min. Then, the fresh human feces were provided by four healthy young volunteers, including two males and two females aged from 20 to 27, without digestive diseases or probiotic or antibiotic intake within the last three months. The feces samples were diluted using normal saline to prepare the solid–liquid mixture (10%, *w*/*v*). The suspension was homogenized and centrifugated (500 rpm) at 4 °C for 5 min to gain the human fecal inoculum. In total, 3.0 mL of feces homogenate was mixed with 27.0 mL of a basic culture medium, containing 300.0 mg HSP80-2 or INL or without any other carbon source (BLK). Then, all groups were anaerobically fermented at 37 °C. Finally, four different fermentation periods (4, 8, 12, and 24 h) of the test samples were collected and stored at a temperature of −80 °C in order to facilitate further analysis.

### 2.9. Determination of SCFAs in the Fecal Fermentation Model of HSP80-2

In brief, sample solutions were subjected to centrifugation to remove impurities at 8000 rpm for 15 min. Subsequently, 400 μL of supernatant was combined with 20 μL of 2-ethylbutyric acid. The mixture was added with 100 μL of 50% H_2_SO_4_ to acidify, and shaken thoroughly. Next, 1000 μL of 2-ethylbutyric acid was added for extraction, and centrifugated at 500 rpm for 10 min at 4 °C. 1.0 μL of the sample supernatants was injected for analysis with the Agilent 7890 series GC system (Agilent Technologies, Palo Alto, CA, USA) with flame ionization detection. The carrier gas was N_2_, and the initial temperature was 100 °C, and rising to 180 °C at a rate of 5 °C/min. The temperatures of injection and the detector were 200 °C and 250 °C [16], respectively.

### 2.10. Analysis of Gut Microbiota during In Vitro Fermentation

After 24 h of fermentation, total genomic DNA was extracted from three groups, including the BLK group without the carbon source, the INL group with the addition of inulin, and the HSP80-2 group with the addition of HSP80-2 immediately using the E.Z.N.A.^®^ Soil DNA Kit (Omega Bio-tek, Norcross, GA, USA) according to the instructions. The quality and concentration of DNA were determined by 1.0% agarose gel electrophoresis and a NanoDrop2000 spectrophotometer (Thermo Scientific, Waltham, MA, USA) and kept at −80 °C prior to further use. The hypervariable region, V3-V4, of the bacterial 16S rRNA gene were amplified with primer pairs 338F (5′-ACTCCTACGGGAGGCAGCAG-3′) and 806R (5′-GGACTACHVGGGTWTCTAAT-3′) by a T100 Thermal Cycler PCR thermocycler (BIO-RAD, California city, CA, USA). Purified amplicons were pooled in equimolar amounts and paired-end sequenced on an Illumina PE300/PE250 platform (Illumina, San Diego, CA, USA) according to the standard protocols by Majorbio Bio-Pharm Technology Co. Ltd. (Shanghai, China). Raw FASTQ 16S rRNA gene sequences were merged by FLASH version 1.2.7. The optimized sequences were clustered into operational taxonomic units (OTUs) using UPARSE 7.1 with 97% sequence similarity level. The metagenomic function was predicted by PICRUSt2 (Phylogenetic Investigation of Communities by Reconstruction of Unobserved States) based on OTU representative sequences. Based on the OTUs information, alpha diversity indices, including Chao1 richness, Shannon index, Ace and Simpson were calculated with Mothur v1.30.1. The similarity among the microbial communities in different samples was determined by principal coordinate analysis (PCoA) based on the Bray–Curtis dissimilarity using Vegan v2.5-3 package. The distance-based redundancy analysis (db-RDA) was performed using Vegan v2.5-3 package.

### 2.11. Statistical Analyses

The data were given as means ± SD (*n* = 3), and were analyzed using a one-way ANOVA followed by the Duncan’s test. If *p* < 0.05, the difference was considered statistically significant. SPSS for Windows, Version 17.0 (SPSS Inc., Chicago, IL, USA) was used for all statistical analyses.

## 3. Results and Discussion

### 3.1. Determination of Mw and Monosaccharide Compositions of HSP80-2

Natural polysaccharides possess a diverse array of biological functions that are linked to their chemical composition and structure, particularly their average molecular weights [17]. As presented in Figure 2a, the HPGPC spectrum demonstrated that HSP80-2 had a solitary symmetrical peak, and the average molecular weight was calculated at 1.38 × 10^4^ Da with the standard curve of y = −0.66 x + 14.38 (R^2^ = 0.9920). Compared with another HS polysaccharide, HSP-3, obtained by 80% ethanol precipitation and eluted with 0.20 mol/L NaCl solution by DEAE-FF column chromatography [18], the average molecular weight of HSP80-2 was significantly smaller than that of HSP-3 (2.64 × 10^4^ Da), revealing the ethanol as the extraction agent had an impact on the distribution of the average molecular weight of polysaccharides, and HSP80-2 prepared by GEP method might exhibit stronger biological activity due to its lower molecular weight [12].

As shown in Figure 2b, the total ion chromatogram (TIC) of HPLC revealed that the composition of HSP80-2 primarily consisted of arabinose, xylose, and glucose with a molar ratio of 4.0: 3.1: 2.4., indicating that HSP80-2 was a heteropolysaccharide. According to the previous studies, HSP-3 contained rich mannose with a content of up to 46%. However, the monosaccharide composition of HSP80-2 was completely different from that of HSP-3. This evidence suggested that although the HSP80-2 and HSP-3 were both originated from the same HS sample, different HSPs had obviously distinct structures, which might be related to different extraction methods.

### 3.2. FT-IR Spectra of HSP80-2

FT-IR spectroscopy was commonly used to reveal typical characteristics of polysaccharide. The FT-IR spectrum of HSP80-2 was shown in Figure 3. The HSP80-2 exhibited a diverse array of stretching vibration characteristic peaks, including a hydroxyl group at around 3354 cm^−1^ and C-H at around 2923 cm^−1^ [19] which explicated that the presence of intermolecular hydrogen bonds within the molecular structure. Moreover, stretching peak at 1651 cm^−1^ and 1429 cm^−1^ belong to COO- [8], which indicated that HSP80-2 had the characteristic peak of a polysaccharide. The strong band distribution of 1100 cm^−1^ to 1000 cm^−1^ indicated the characteristic of C-O-C and C-O-H bonds, which evidenced the presence of a pyranose form of sugars [20]. The absorption peak at 812 cm^−1^ was attributed to the presence of α-pyran sugar in the HSP80-2 compound [21]. It was worth noting that HSP80-2 had a similar functional group structure to HSP-3 which both had characteristic absorption peaks of polysaccharides [20].

### 3.3. Morphological Properties of HSP80-2

AFM technique can image single molecules and aggregates of polysaccharides, obtain quantitative information such as the diameter and length of single molecules, as well as morphological characteristics of molecular aggregates, and has been widely applied in the study of polysaccharides [22]. The planar images (Figure 4a,c) showed that HSP80-2 had a distinct morphology characterized by a short branch chain structure, and the 3D images (Figure 4b,d) showed the rough surfaces of HSP80-2. The molecular height of HSP80-2 was in the range of 1.09–2.98 nm, and the diameter of the dispersed particles was 66.12–107.15 nm. Interestingly, HSP-3 formed a nonlinear spherical structure due to the highly intertwined sugar chains which was totally different from the image structure characterization of HSP80-2. It suggested that the HSP80-2 had a more complex polysaccharide structure.

The molecular morphology of polysaccharides was investigated by SEM under 500× and 1000× magnifications. Figure 4e shows that the surface morphology of HSP80-2 was a flaky structure and uneven size. As shown in Figure 4f, HSP80-2 had a rather integrated and smooth surface at a high magnification. This could be owing to HSP80-2 having a low degree of branched and short strain structures [14].

### 3.4. Changes in Gut Microbiota Population

It is widely recognized that the gut microbiota has a crucial role in both immune system functioning and energy metabolism [23]. Polysaccharides have been reported to elicit bioactive effects via the gut microbiota and its metabolites [24]. Consequently, the investigation of the correlation between polysaccharides and gut microbiota has promise for illness prevention and health promotion.

In the present study, a high-throughput sequencing study was conducted on the samples obtained from the INL, BLK, and HSP80-2 groups following 24 h fermentation to investigate the impact of HSP80-2 on the composition of gut microbiota. Table 1 displayed the alpha diversity metrics of the community, including the community richness (Chao1 and ACE) and the distributional diversity of the community (Simpson and Shannon). The study demonstrated that the BLK group had a higher functional role in preserving community abundance and diversity compared to the HSP80-2 and INL groups, the same results were obtained in the polysaccharides extracted from *Paecilomyces cicadae* TJJ1213 in the *in vitro* fermentation [25]. The reason might be that the intestinal microorganisms use inulin and polysaccharide to produce SCFAs, which could inhibit the growth of harmful bacterial communities and decrease the abundance of intestinal microorganisms [26].

As shown in Figure 5a, principal component 1 (PC1) accounted for 52.97% of the variance observed among the three groups, and the principal component 2 (PC2) accounted for 45.12% of the variation, which suggested that the microbiota shifted in both HSP80-2 and INL groups comparing to the BLK group. The obtained results showed that the gut microbiota composition and structure of HSP80-2 and INL were completely different from that of BLK group, and further demonstrated that HSP80-2 and INL could change the composition of gut microbiota during a simulated colonic fermentation.

The relative bacterial community abundances of three groups at the phylum level were shown in Figure 5b, and the results indicated the bacterial community of INL, BLK, and HSP80-2 was mainly containing *Firmicutes*, *Bacteroidetes*, *Fusobacteria* and *Proteobacteria*. After 24 h of fermentation, the microbiota composition of HSP80-2 obviously differed from that in the BLK group and INL group. The HSP80-2 group exhibited a notable elevation in the abundance of *Bacteroidetes*, while the level of *Firmicutes* decreased compared with the BLK and INL groups. Previous study showed that *Bacteroidetes* could utilize more types of polysaccharides than *Firmicutes*, as *Bacteroidetes* can encode more polysaccharide degrading enzymes [27]. *Bacteroides* was known to inhabit the distal colon and possess the ability to enzymatically break down indigestible polysaccharides, resulting in the production of SCFAs. This capability was facilitated by the presence of many carbohydrate enzymes, including glycosidases and polysaccharide lyases [28]. In addition, the relative abundance of *Proteobacteria* in the HSP80-2 and INL groups were significantly lower compared with that of the BLK group. As reported, *Proteobacterias* was the largest phylum of bacteria and includes many pathogens, which were recognized as a significant factor in the development of intestinal microecological problems, leading to potential nutritional and metabolic disturbances in the host. Additionally, several bacterial genera within this phylum have been associated with immune-related disorders [29]. These results indicated that HSP80-2 could regulate the structure of bacterial communities to promote health of host.

In addition, the genus classification information was represented in Figure 5c. The BLK group was mainly composed of *Fusobacterium*, *Clostridium_sensu_stricto_18*, *Flavonifractor*, and *Escherichia-Shigella*, and clearly, the probiotics almost disappeared. In contrast to the BLK group and INL, the HSP80-2 group exhibited a higher abundance of *Bacteroides*, which was one of the most abundant human intestinal microbiotas and played an important role in the development and maintenance of healthy intestinal microbiota. *Bacteroides* were increasingly used as a model organism to study the physiology and function of intestinal symbiotic microorganisms, such as polysaccharide utilization, pathogenicity, bile acid metabolism, and the ecology and evolution of intestinal microbiota. Furthermore, the HSP80-2 group demonstrated elevated quantities of *Phascolarctobacterium* and *Lachnoclostridium* compared to the BLK group, similar results were also observed in polysaccharides isolated from *agaricus bisporus* [30]. The presence of a higher quantity of *Phascolarctobacterium* in the gastrointestinal microbiota of adult individuals had been found to promote weight reduction. Additionally, the presence of Dialister genes that encode enzymes involved in carbohydrate metabolism among gut microorganisms also play a role in facilitating weight loss [31]. The results were also observed in the INL group with increased *Phascolarctobacterium* when compared with the BLK group. Meanwhile, a reduction in the abundance of *Fusobacterium* was noted in both the INL and the HSP80-2 compared to the BLK group. Previous research had demonstrated a correlation between the gut microbiome and the development of colorectal cancer (CRC). Specifically, an excessive presence of *Fusobacterium* in the colon had consistently emerged as a reliable indicator in this context [32]. In summary, the HSP80-2 could regulate gut microbiota by increasing beneficial microbiota and reducing the abundance of harmful microbiota. Meanwhile, Figure 5d displayed a heat map that illustrated the relative abundance of the top 50 bacterial genera in the BLK, HSP80-2, and INL groups. In the HSP80-2 group, there was an observed rise in the abundance of profitable bacterial genera, including *Lachnoclostridium*, *Parabacteroides*, *Phascolarctobacterium*, and *Bacteroides* compared to the BLK groups. This result further confirmed that HSP80-2 could increase the abundance of beneficial microbiota, suggesting HSP80-2 displayed the potential to support and preserve intestinal health.

### 3.5. Prediction of HSP80-2 on the Metabolic Pathways of the Gut Microbiota by KEGG Analysis

The PICRUSt2 is used to predict the metagenomic pathways based on the Kyoto Encyclopedia of Genes and Genomes (KEGG) database. In the present study, a total of 44 KEGG pathways on level 2 were obtained, of which 41 exhibited significant differences compared to BLK (Appendix A). The level 2 KEGG pathways, with the top 14 relative abundance, were displayed at Figure 6a, which were mainly related to global and overview maps, carbohydrate metabolism, and glycan biosynthesis and metabolism. To further investigate metabolic pathways, KEGG pathway analysis on level 3 was carried out and 320 pathways were found, of which 256 pathways had significant differences between the BLK and HSP80-2 groups (Appendix A). The results indicate that carbohydrate metabolism-related pathways, including amino sugar and nucleotide sugar metabolism (Ko00520), galactose metabolism (ko00052), citrate cycle (TCA cycle) (ko00020), and other glycan degradation (ko00511) were enriched in the HSP80-2 group (Appendix A). In addition, the carbohydrates could regulate the production of amino acids by affecting the gut microbiota [33]. The metabolic pathways of most amino acids with HSP80-2 treatment were enriched (Appendix A), such as glycine, serine and threonine metabolism (ko00260), alanine, aspartate and glutamate metabolism (ko00250), lysine biosynthesis (ko00300), arginine and proline metabolism (ko00330), and valine, leucine and isoleucine biosynthesis (ko00290). These results suggested that HSP80-2 could be utilized by the gut microbiota and affect the production of amino acids. Additionally, the Level 3 KEGG pathways with the top 14 relative abundance were shown in Figure 6b, which were mainly related to metabolic pathways (ko01100), biosynthesis of secondary metabolites (ko01110), and microbial metabolism in diverse environments. Similar to previous research results [34], the pathways related to metabolic pathways (ko01100), biosynthesis of secondary metabolites (ko01110), biosynthesis of amino acids (ko01230), and amino sugar and nucleotide sugar metabolism (ko00520) were significantly increased in the HSP80-2 group compared with the BLK group, which further indicated that HSP80-2 could improve the metabolic pathway of amino acids.

In order to investigate the relationship between the main pathways and the gut microbiota, the Spearman correlation heatmap in Figure 6c showed the correlation between 14 metabolic pathways (the Level 3 KEGG pathways with the top 14 relative abundance) and 15 genera (the genera with top 15 relative abundance). The results indicate that *Bacteroides*, *Lachnoclostridium*, and *Parabacteroides* were significantly positively correlated with multiple metabolic pathways, such as Biosynthesis of amino acids (ko01230), Carbon metabolism (ko01200), Amino sugar and nucleotide sugar metabolism (ko00520), Purine metabolism (ko00230), and Pyrimidine metabolism (ko00240). It was worth noting that *Escherichia-Shigella*, *Clostridium_sensu_stricto_1*, and *Clostridium_sensu_stricto_18* was significantly negatively correlated with the aforementioned metabolic pathways. Based on the biological functions of these genera, we speculated that *Bacteroides*, *Lachnoclostridium*, and *Parabacteroides* play an active role in these metabolic processes.

### 3.6. SCFAs Production during In Vitro Fermentation

SCFAs are one of the most prominent by-products of polysaccharides fermentation by intestinal microbiota, which play a significant and beneficial function in intestinal epithelial cell regulation [24]. Inulin (INL) was a widely recognized type of prebiotic that was frequently employed as a positive control in *in vitro* fermentation investigations. The SCFAs levels during different fermentation periods of HSP80-2, INL, and BLK were detected by GC technique, respectively, and the results were shown in Table 2. With the increase of fermentation time in the BLK, INL, and HSP80-2 groups, the contents of total SCFAs gradually increased. Similar results were observed in the fermentation experiments of *aloe* polysaccharide [35], *Artemisia sphaerocephala* Krasch [36], and *Siraitia grosvenorii* polysaccharide [37]. It was worth noting that HSP80-2 produced substantially more SCFAs than BLK and INL at any time point. After 24 h of fermentation, total SCFAs concentration in the HSP80-2 group grew to 47.79 ± 2.71 mmol/L, significantly higher than in the BLK (14.86 ± 1.40 mmol/L) and INL (31.48 ± 1.40 mmol/L) groups. These results indicated that HSP80-2 and INL regulated the gut microenvironment might by producing SCFAs progressively, which was consistent with the fermentation results of extracting polysaccharides from *Ziziphus Jujuba* [38].

In fact, the acetic, propionic, and n-butyric were the primary SCFAs in HSP80-2. After fermentation for 24 h, the content of acetic acid and propionic acid in the HSP80-2 group were determined to be 25.63 ± 2.91 mmol/L and 13.12 ± 2.16, respectively, which was significantly higher than those of the INL (15.37 ± 0.69 mmol/L and 3.98 ± 0.25 mmol/L) group and BLK group (8.92 ± 1.01 mmol/L and 1.38 ± 0.21 mmol/L). Notably, the levels of acetic acid and propionic acid in the HSP80-2 group were significantly higher than those in BLK and INL at all fermentation periods. Acetate produced via acetyl-CoA and Wood-Ljungdahl pathways, which was the most abundant SCFAs of all substrates. At the same time, acetic acid was an important energy source for gut microbiota and liver cells, which could be utilized in brain, heart, and to inhibit enteropathogens [39]. The liver could absorb propionic acid to inhibit cholesterol synthesis and lower serum cholesterol levels, propionic acid was also a significant molecule in satiety signaling for interactions with FFAR2/3 [40]. Meanwhile, after fermentation for 24 h, the butyric acid in the HSP80-2 group was 8.74 ± 1.51 mmol/L, which was 2.31-fold of it in BLK (3.77 ± 0.33 mmol/L) but slightly lower than that in the INL group (11.96 ± 0.85 mmol/L). The phenomenon observed in this study was similar to the report by Wen et al. in the fermentation of *Sparassis crispa* polysaccharide [41]. Butyric acid had significant implications in the control of host genes, cellular development, and cellular death. Moreover, it was considered to be a primary energy source for epithelial cells [4]. These results collectively indicated that HSP80-2 had positive effects as inulin on promoting the production of SCFAs, thus, protecting intestinal health as a potential probiotic.

In order to link changes in microbiota composition with the different fermentation conditions and different SCFAs, the redundancy analysis at the phylum level and genus level were performed. At the level of phylum as represented in Figure 7a, the content of the SCFAs were closely related to the relative abundance of *Bacteroides*. At the level of genera as shown in Figure 7b, the production of acetic acid and propionic acid was closely related to the growth of *Bacteroides*, *Lachnoclostridium,* and *Parabacteroides* which were positively regulated by HSP80-2.

## 4. Conclusions

In conclusion, this study characterized a new polysaccharide from HS by the GEP method, and investigated its prebiotic activity by the *in vitro* fermentation model, which would provide a valuable reference for increasing the added value of HS, the by-product of Chinese Baijiu. Specifically, a novel heteropolysaccharide, HSP80-2, with an average molecular weight of 13.8 kDa was successfully extracted, which was mainly composed of arabinose, xylose, and glucose with a molar ratio of 4.0: 3.1: 2.4. In addition, HSP80-2 had a smooth surface fragmented structure and a short branch chain structure.

Furthermore, HSP80-2 exhibited prebiotic activity, which was attributed to the remarkable increasing contents of SCFAs, including acetic acid, butyric acid, and propionic acid, and the significant enhancement on the abundance of beneficial microbiota such as *Bacteroides* and *Phascolarctobacterium* after 24 h of fermentation in vitro. Also, it was worth noting that the production of acetic acid and propionic acid was closely related to the growth of *Lachnoclostridium* and *Parabacteroides*. Furthermore, the biosynthesis of amino acids was positively correlated with *Bacteroides*, *Lachnoclostridium*, and *Parabacteroides*, which were closely related to the treatment of HSP80-2. Although positive results had been achieved, the changes in HSP80-2 during *in vitro* fermentation and other products after fermentation were still worthy of further investigation. Meanwhile, as the result of the *in vitro* fermentation model cannot fully represent the interaction between HSP80-2 and gut microbiota, further research is needed to verify.

## Figures and Tables

**Figure 1 foods-12-04406-f001:**
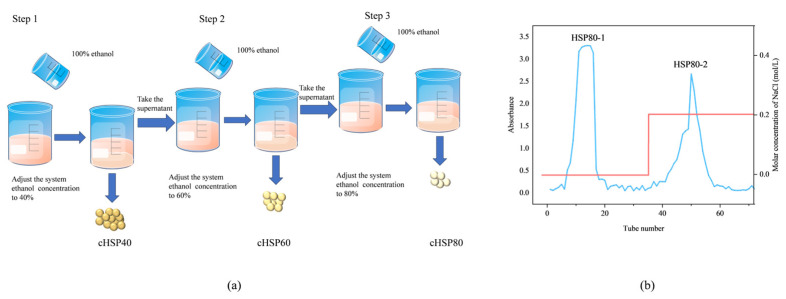
(**a**) The schematic diagram of the processing process of GEP method, and (**b**) the Stepwise elution curve of cHSP80 on a DEAE-FF column, in which HSP80-1 and HSP80-2 represented the fractions eluted with water and 0.20 mol/L of NaCl solution, respectively.

**Figure 2 foods-12-04406-f002:**
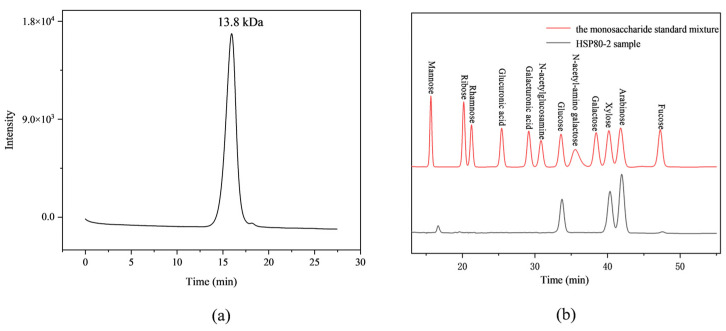
(**a**) HPGPC profile of HSP80-2 fractions on TSK gel GMPWXL column, and (**b**) HPLC analysis of the monosaccharide standards and HSP80-2 sample.

**Figure 3 foods-12-04406-f003:**
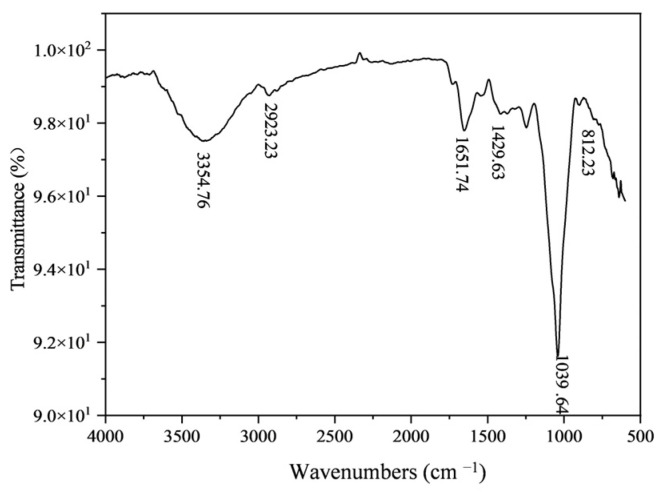
FT-IR spectrum analysis of HSP80-2.

**Figure 4 foods-12-04406-f004:**
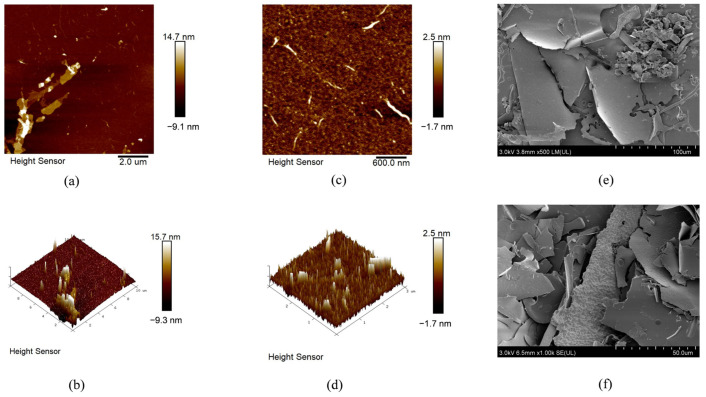
Molecular morphology of HSP80-2 was observed under AFM ((**a**,**c**): planar image, (**b**,**d**): cubic image; (**a**,**b**): 2.0 μm, (**c**,**d**): 600 nm), and SEM image ((**e**) 500× and (**f**) 1000×).

**Figure 5 foods-12-04406-f005:**
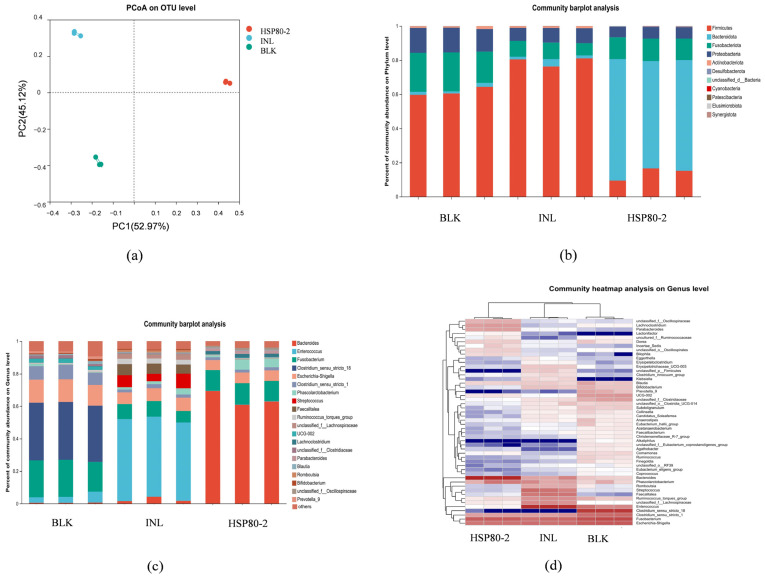
Principal component analysis (**a**) of gut microbiota. The relative abundance of bacterial community at the phylum level (**b**) and the genus level (**c**), and the heatmap analysis of the relative abundance of bacterial community at the genus level (**d**).

**Figure 6 foods-12-04406-f006:**
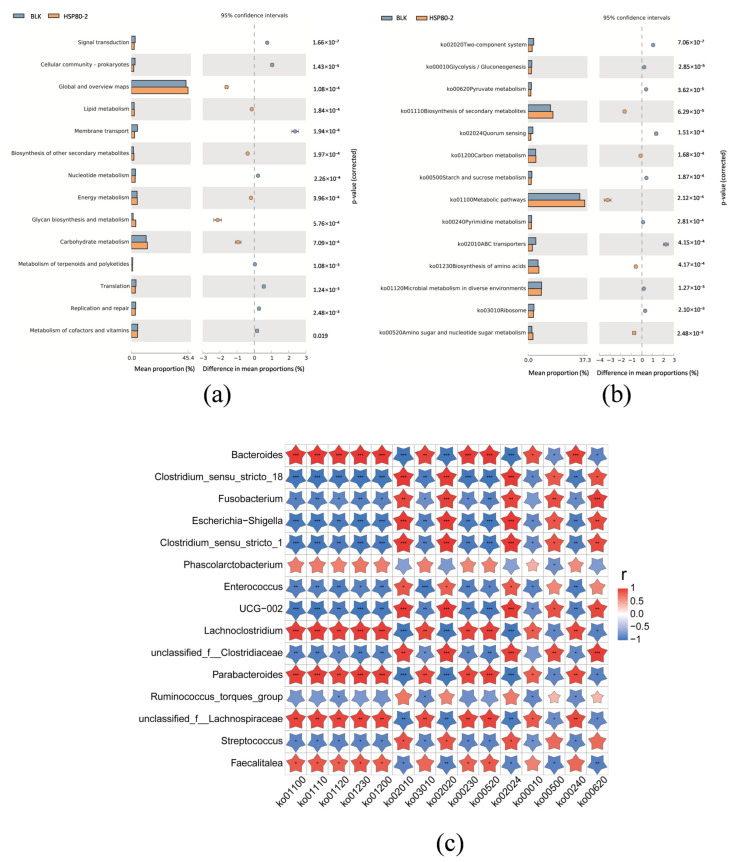
Difference of PICRUSt function prediction based on KEGG database after PHSP80-2 fermentation: (**a**) Relative abundance of the top 14 metabolic pathways base on KEGG categories at Level 2. (**b**) Relative abundances of the top 14 metabolic pathways base on KEGG categories at Level 3. (**c**) Heatmap showed Spearman correlations between 14 different metabolic pathways in Level 3 and 15 different genera, * *p* < 0.05, ** *p* < 0.01 and *** *p* < 0.001 denoted statistical significance between bacterial taxa and metabolic pathways.

**Figure 7 foods-12-04406-f007:**
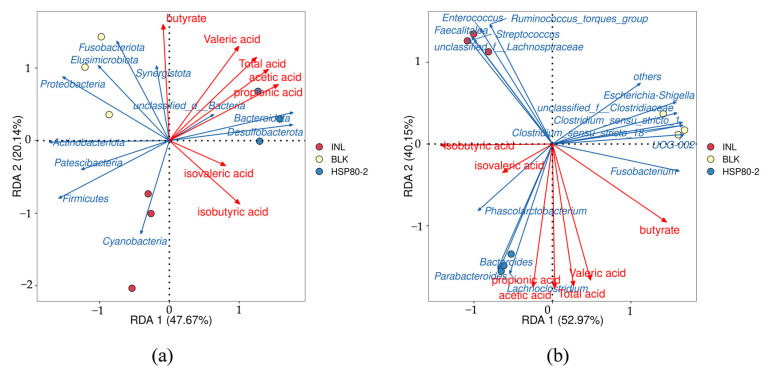
The correlation analysis between SCFAs and microbial diversity at the phylum (**a**) and genus (**b**) level.

**Table 1 foods-12-04406-t001:** Alpha diversity of samples among different groups.

Groups	Index			
	Ace	Shannon	Chao 1	Simpson
BLK	313.65 ± 14.44 ^a^	2.32 ± 0.13 ^a^	312.23 ± 22.93 ^a^	0.20 ± 0.01 ^b^
INL	282.2 ± 44.49 ^ab^	2.14 ± 0.06 ^a^	270.03 ± 37.24 ^ab^	0.27 ± 0.01 ^a^
HSP80-2	234.76 ± 14.21 ^b^	2.33 ± 0.13 ^a^	236.93 ± 21.39 ^b^	0.20 ± 0.03 ^b^

^a, b^ Significant differences (*p* < 0.05) are expressed with superscript letters within a row for each parameter (*n* = 3).

**Table 2 foods-12-04406-t002:** Changes in concentrations of SCFAs produced at different fermentation time.

Sample Group	Fermentation Time (h)	Short-Chain Fatty Acids Content (mmol/L)						
		Acetic Acid	Propionic Acid	i-Butyric Acid	N-Butyric Acid	i-Valeric Acid	n-Valeric Acid	Total
BLK	0	ND	ND	ND	ND	ND	ND	ND
	4	0.78 ± 0.07 ^f^	0.18 ± 0.02 ^d^	0.057 ± 0.01 ^c^	0.33 ± 0.03 ^f^	0.036 ± 0.008 ^c^	0.016 ± 0.005 ^e^	1.41 ± 0.10 ^f^
	8	1.42 ± 0.11 ^f^	0.23 ± 0.03 ^d^	0.066 ± 0.02 ^c^	0.52 ± 0.02 ^f^	0.046 ± 0.009 ^c^	0.017 ± 0.003 ^e^	2.37 ± 0.14 ^f^
	12	2.34 ± 0.10 ^f^	0.48 ± 0.02 ^d^	0.075 ± 0.01 ^b,c^	0.56 ± 0.03 ^f^	0.046 ± 0.008 ^b,c^	0.020 ± 0.005 ^c,d^	3.51 ± 0.07 ^f^
	24	8.92 ± 1.01 ^d^	1.38 ± 0.21 ^c^	0.12 ± 0.02 ^a,b^	3.77 ± 0.33 ^d^	0.049 ± 0.008 ^b,c^	0.019 ± 0.004 ^c,d^	14.86 ± 1.40 ^d^
HSP80-2	0	ND	ND	ND	ND	ND	ND	ND
	4	1.62 ± 0.13 ^f^	0.24 ± 0.01 ^d^	0.058 ± 0.02 ^c^	0.36 ± 0.16 ^f^	0.032 ± 0.003 ^c^	0.015 ± 0.003 ^e^	2.33 ± 0.34 ^f^
	8	9.61 ± 1.01 ^d^	1.36 ± 0.21 ^c^	0.091 ± 0.01 ^a,b,c^	2.30 ± 0.39 ^e^	0.047 ± 0.007 ^b,c^	0.026 ± 0.003 ^c,d^	13.16 ± 1.69 ^d^
	12	13.21 ± 0.78 ^c^	4.64 ± 0.07 ^c^	0.10 ± 0.03 ^a,b,c^	3.50 ± 0.48 ^d^	0.062 ± 0.007 ^b^	0.029 ± 0.005 ^c^	21.53 ± 1.05 ^c^
	24	25.63 ± 2.91 ^a^	13.12 ± 2.16 ^a^	0.13 ± 0.07 ^a^	8.74 ± 1.51 ^d^	0.081 ± 0.007 ^a^	0.075 ± 0.006 ^a^	47.79 ± 2.71 ^a^
INL	0	ND	ND	ND	ND	ND	ND	ND
	4	0.9 ± 0.43 ^f^	0.21 ± 0.05 ^d^	0.058 ± 0.003 ^c^	0.35 ± 0.03 ^f^	0.0408 ± 0.006 ^c^	0.019 ± 0.005 ^c,d^	1.56 ± 0.07 ^f^
	8	4.52 ± 0.08 ^e^	0.89 ± 0.08 ^d^	0.059 ± 0.01 ^c^	1.05 ± 0.03 ^c^	0.046 ± 0.009 ^b,c^	0.019 ± 0.001 ^c,d^	6.58 ± 0.10 ^e^
	12	8.23 ± 0.31 ^d^	1.01 ± 0.04 ^c^	0.064 ± 0.005 ^c^	4.65 ± 0.06 ^b^	0.050 ± 0.004 ^b,c^	0.025 ± 0.001 ^c,d^	14.05 ± 0.40 ^d^
	24	15.37 ± 0.69 ^b^	3.98 ± 0.25 ^b^	0.067 ± 0.01 ^c^	11.96 ± 0.85 ^a^	0.048 ± 0.011 ^b,c^	0.026 ± 0.02 ^b^	31.48 ± 1.75 ^b^

^a–f^ Significant differences (*p* < 0.05) are expressed with superscript letters within a row for each parameter. (*n* = 3). ND: Not detected.

## Data Availability

The data presented in this study are available on request from the corresponding author.

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
