# Peer review of "The Prebiotic Activity of a Novel Polysaccharide Extracted from Huangshui by Fecal Fermentation In Vitro"

_foods, 2023, doi:10.3390/foods12244406_

Round 1

Reviewer 1 Report

Comments and Suggestions for Authors

In this study, Li et al. investigated the effects of a novel polysaccharide extracted from Huangshui on fecal fermentation in vitro. The manuscript is well-written, but there are still a few issues. Here are some comments on this paper:

1.       Section 2.4, it is proposed that authors provide a detailed method for the processing of gut microbiome data, such as processing and visualization software, and database.

2.      The title of “3. Results” should be “3. Results and discussion”.

3.      Figures 4 e and f are identical, please verify that the figures are correct.

4.      Table 1 and Figure 5 (a) and (b) both show results on the diversity of the intestinal microbiome, which appears to be a bit redundant, and it is recommended to retain Table 1.

5.      During the in vitro fermentation, were there results of changes in total carbohydrates and monosaccharides?

Reviewer 2 Report

Comments and Suggestions for Authors

he prebiotic activity of a novel polysaccharide extracted from  Huangshui by fecal fermentation in vitro

Mei Li1 , Jian Su2, Jihong Wu1, Dong Zhao2 , Mingquan Huang1 , Yanping Lu2, Jia Zheng2 and Hehe Li1 

This research covers tasks according to the fermentation, industrial biotechnology, and food technology. This study showed, that a novel polysaccharide, HSP80-2, with an average molecular weight of 13.8 kDa was successfully isolated by gradient ethanol precipitation method from Huangshui, the by-product of Chinese Baijiu. HSP80-2 was utilized by gut microbiota, and significantly regulated the composition and abundance of beneficial microbiota such as Phascolarctobacterium, Parabacteroides, and Bacteroides. HSP80-2 also showed prebiotic activity evaluated by in vitro fermentation.

Some comments to the authors:

1)     It is not clear if „it is probiotic or prebiotic activity of polysaccharides“ in the paragraph of 36.

2)     It should be written 20 µL in the paragraph of 167.

3)     To correct „ethyl ethe“ in the paragraph of 169.

4)     It is difficult to see the „x and y axes“ (very small size) of the Figures 5, 6.

5)     It should be written „and the HSP80-2“ in the paragraph 325.

6)     Please write more in detail about which Figure the authors are writing: „In order to investigate the relationship between the main pathways and the gut microbiota, the Spearman correlation heatmap in Figure“ in the paragraph of 367.  

7)     It is not clear what fermentation volume was used in the studies.

8)     It is not clear what is means „aloe polysaccharide“ in the paragraph of 389.

Reviewer 3 Report

Comments and Suggestions for Authors

Recommendation: Major

The manuscript The prebiotic activity of a novel polysaccharide extracted from Huangshui by fecal fermentation in vitro was reasonable and technically sound.

Comments to the Author:

Below are some suggestions.

Point 1: Briefly explain the preparation method of samples for SEM analysis.

Point 2: Was ethics committee permission or approval obtained for the human feces study?

Point 3: The figures cannot be understood because their DPI is not good. It should be presented in high resolution.

Point 4: Add explanations of the abbreviations below the tables.

Point 5: Figure 5 and Figure 6 resolutions are very poor. It is impossible to evaluate. The heat map is not especially understandable. Some figures should be presented separately and the size and resolution should be increased. It has become impossible to evaluate some sections.

Point 6: What is the statistical analysis performed in Figure 7? What do you mean by RDA (redundancy analysis)? With which program was RDA made? Does this issue need to be stated clearly? Comments and discussions should be enhanced.

Point 7: I recommend that you make suggestions for future studies in the concluding sentence.

Comments on the Quality of English Language

Minor revisions to grammar are needed.

Round 2

Reviewer 3 Report

Comments and Suggestions for Authors

The authors made the necessary revisions.